# The Impact of COVID-19 on the Emotion of People Living with and without HIV

Joanne Lusher [1,2,*], Roberto Ariel Abeldaño Zuñiga [1,3], Jorma I. Virtanen [1,4], Passent Ellakany [1,5], Muhammad Abrar Yousaf [1,6], Bamidele Emmanuel Osamika [1,7], Balgis Gaffar [1,8], Folake Barakat Lawal [1,9], Zumama Khalid [1,10], Nourhan M. Aly [1,11], Annie Lu Nguyen [1,12] and Morenike Oluwatoyin Folayan [1,13]

1 Mental Health and Wellness Study Group, Obafemi Awolowo University, Ile-Ife 22005, Nigeria
2 Provost's Group, Regent's University London, London NW1 4NS, UK
3 Postgraduate Department, University of Sierra Sur, Oaxaca 70800, Mexico
4 Faculty of Medicine, University of Turku, 20014 Turku, Finland
5 Department of Substitutive Dental Sciences, College of Dentistry, Imam Abdulrahman Bin Faisal University, Dammam 32210, Saudi Arabia
6 Department of Biology, Faculty of Science and Technology, Virtual University of Pakistan, Lahore 54000, Pakistan
7 Department of Psychology and Institute for the Environment and Sustainability, Miami University, Oxford, OH 45056, USA
8 Department of Preventive Dental Sciences, College of Dentistry, Imam Abdulrahman Bin Faisal University, Dammam 34212, Saudi Arabia
9 Department of Periodontology and Community Dentistry, University of Ibadan, Ibadan 200212, Nigeria
10 Department of Health Sciences, University of Genova, 16132 Genova, Italy
11 Department of Pediatric Dentistry and Dental Public Health, Faculty of Dentistry, Alexandria University, Alexandria 31773, Egypt
12 Department of Family Medicine, Keck School of Medicine, University of Southern California, Los Angeles, CA 91803, USA
13 Department of Child Dental Health, Obafemi Awolowo University, Ile-Ife 22005, Nigeria
* Correspondence: lusherj@regents.ac.uk

**Abstract:** The COVID-19 pandemic is a source of mental stress, particularly for special populations. The present study identified the associations between emotional distress and HIV status among adults in 152 countries during the first wave of the pandemic. This was a cross-sectional study that gathered data via an online survey carried out between July and December 2020. The dependent variable was emotional distress (frustration/boredom, anxiety, depression, loneliness, anger, or grief/feeling of loss), and the independent variable was HIV status (positive or negative). The confounding factors were sociodemographic variables (age, sex, education level, and employment status). Multivariable logistic regression analyses were performed to assess the associations between the independent and dependent variables after adjusting for confounders. Of the 16,866 participants, 905 (5.4%) reported living with HIV. Of these, 188 (20.8%) felt frustrated/bored, 238 (26.3%) anxious, 160 (17.7%) depressed, 148 (16.4%) lonely, 84 (9.3%) angry, and 53 (5.9%) grief/a sense of loss. Individuals living with HIV had higher odds of feeling anxious (AOR:1.64), depressed (AOR:1.80), and lonely (AOR:1.35) when compared to people living without HIV. This study reveals that the COVID-19 pandemic can exacerbate emotional stress for those living with HIV, and the system in which COVID-19 impacts emotional health among different sociodemographic groups introduces further complexities regarding this observed effect.

**Keywords:** COVID-19 pandemic; HIV; emotion; distress; frustration; boredom; anxiety; depression; loneliness; anger

## 1. Introduction

The COVID-19 pandemic has been a source of mental health challenges because of its effect on multiple aspects of life at the individual, household, and national levels [1].

The pandemic is also associated with social isolation [2]. The combination of financial instability, social isolation, fear of contracting COVID-19, and uncertainty surrounding the future are factors that have previously been shown to cause substantial emotional stress [3]. Moreover, these factors have compounded feelings of frustration [4,5], boredom [6], loneliness [7–10], anger [11], grief [12], loss [13], anxiety, and depression [14,15] for individuals on a worldwide scale, and, in particular, for those living with chronic conditions [16].

For special populations, such as those living with HIV, frustration [16], boredom [17,18], loneliness [19], anger [20], grief [21], feelings of loss [22], anxiety [23], and depression [24–26] have been commonly reported. This range of emotions is known to be triggered by pressures related to food, financial, and housing insecurities, which have been further impacted by the COVID-19 pandemic lockdowns and restrictions [27,28]. These same issues produced emotional distress among people living with HIV before the pandemic [29–32], as they have had to deal with the negative emotions resulting from a diagnosis of HIV infection because of the harmful ideas and beliefs that exist about HIV. These harmful ideas create a stigma in others' minds and shame in the infected individual as an emotional response to this stigma, which is based on the negative evaluation of someone who has fallen short of some societal standard [33]. The emotional response to shame is a physiological change triggered by causal stimuli in the environment [34]. The cognitive and behavioral manifestations of shame are negative self-conscious emotions [35], such as anxiety and depression [36]. The emotional responses to the pandemic by people living with HIV have been blunted by their prior experiences of emotionally stressful events [37,38]. The pandemic may, rather, evoke the usual emotions associated with living with HIV as it echoes the dawn of the AIDS era [39].

People not living with HIV may, however, experience basic emotions in response to the pandemic-related stress. The COVID-19 pandemic is a highly negative event that evokes a stressful context perceived as uncertainty or threat. In this condition, information is processed efficiently and with the priority to mitigate the negative effects of the stimulus resulting in the simplicity of information processing. The result is an increase in unpleasant emotions along with a decline in pleasant emotions [40]. These basic unpleasant emotions are external expressions of stereotypical behaviors in response to internal states that are modulated by neuromodulators [41]. The external expressions include anger, frustration, guilt, loneliness, fear, boredom, and or loss [4–13].

In view of the different neurological pathways for the experience of emotions by people living with and without HIV during the COVID-19 pandemic, it may be possible that the forms of emotions expressed during the pandemic may differ between the two populations. The present report explored this notion by investigating the association between emotional stress and HIV status during the first wave of the COVID-19 pandemic.

## 2. Materials and Methods

Ethical approval for the study was obtained from the Human Research Ethics Committee at the Institute of Public Health of the Obafemi Awolowo University Ile-Ife, Nigeria (HREC No: IPHOAU/12/1557), Brazil (CAAE N° 38423820.2.0000.0010), India (D-1791-uz and D-1790-uz), Saudi Arabia (CODJU-2006F), and the United Kingdom (13283/10570) for the conduct of the primary study. Study participants checked a box to indicate consent before participating in the online survey.

### 2.1. Study Design and Participants

This was a secondary analysis of data extracted from a primary study on mental health and wellness that recruited 21,106 participants from 152 countries between July and December 2020 through an online survey. Participation was open to anyone 18 years and older. There were no exclusion criteria.

## 2.2. Sample Size

The sample size was computed based on the highest global prevalence of mental health disorders in 2019. The pre-survey minimum sample size for this study was based on the estimated prevalence of the most common global mental health disorder in 2019 (3.94% for anxiety disorder) [42]; the desired precision of the estimate was 0.05, and the confidence level was 95% for an infinite population size [43]. Thus, the sample size was set at 59 valid participants from each of the 193 member states of the United Nations. The sample size was increased by 10% to allow for the risk of missing responses [44]. Online data collection was carried out in view of the restrictions in place during the first wave of the COVID-19 pandemic. Based on statistical modeling, the sample size was adequate when there was a minimum of 10 participants with complete responses per each of the dependent variables for the study as this enables the performance of regression analyses with a minimum probability level (*p*-value) of 0.05 [45].

This study formed part of a larger project, and the methods and procedures have been previously reported and published elsewhere [46–48]. The main project adopted non-probability sampling with recruitment being driven by 45 members of the MEHEWE study group (www.mehewe.org; accessed on 1 January 2021). Members shared a survey link with their contacts around the world using various social media platforms (Facebook, Twitter, and Instagram), network email lists, and groups.

## 2.3. Data Collection Tool

The data collection tool used in this study was validated using both quantitative and qualitative assessments [48]. The instrument was first developed in English and then translated into French, Spanish, Arabic, and Portuguese. The translated tools were translated back into English to ensure they retained their meaning. The overall content validation index for the study questionnaire was 0.83. The dimensionality and reliability of the tool were also assessed. Further details surrounding the validation of this data collection tool are available elsewhere [48].

## 2.4. Study Procedure

The data were collected anonymously. The privacy of participants and the confidentiality of the information they provide were also protected by decoupling the IP addresses from the questionnaire at the end of the online survey. In addition, the survey did not install any tracker cookies on the device of the respondents. The data were collected using SurveyMonkey® (Survey Monkey, Momentive Inc.: San Mateo, CA, USA), which is a secure SSL-encrypted connection link. The data in transit (while responding online) were encrypted using secure TLS cryptographic protocols. This collection tool was certified in compliance with the E.U.-U.S. Privacy Shield Framework and Swiss-U.S. Privacy Shield.

## 2.5. Data Analyses

The dependent variable used for the analyses was emotional distress (frustration or boredom, anxiety, depression, loneliness, anger, and grief/feeling of loss). Participants were required to indicate whether they experienced any of the listed forms of emotional distress during the pandemic by checking a box against each emotion. Participants who did not check a response were categorized as not having experienced emotional distress during the pandemic. The content validation for the section of the questionnaire containing details about emotional status during the pandemic was 0.90. The test–retest reliability score ranged from 0.09 to 0.91. The discriminant measures had a mean variance of explanation of 15.9% and an overall variance explained by two dimensions of 31.8%. The strongest discriminant measures were anxiety (0.37), frustration (0.34), loneliness (0.34), and depression (0.32).

HIV status was treated as an independent variable. Participants indicated their HIV status by checking a list of 27 medical ailments. A tick in the checkbox for HIV was an indication that the individual was living with HIV. The list of medical ailments was adopted

from previous work [49]. The content validation for the section of the questionnaire that contained details on the emotional status during the pandemic was 0.71 [48].

Confounders in this study were age at last birthday, sex at birth (male, female), level of completed education (none, primary, secondary, or college/university), employment status (retiree, student, employed, or unemployed), and country income level. Information about the country income level was obtained from publicly available data from the World Bank Data Bank [50]. Based on income level, countries were classified into low-income countries (LIC) with a gross national income (GNI) per capita of $\leq$1035 USD in 2019, lower–middle-income countries (LMIC) with a GNI between 1036 and 4045 USD, upper–middle-income countries (UMIC) with a GNI between 4046 and 12,535 USD, and high-income countries (HIC) with a GNI of $\geq$12,536 USD.

Data on 16,866 participants with complete dependent, independent, and confounding variables were extracted for analyses. The raw data were downloaded, imported, and cleaned in SPSS version 23.0 (IBM Corp., Armonk, NY, USA) for analyses. Multivariate logistic regression analyses were conducted to determine associations between the dependent and independent variables after adjusting for confounders. Adjusted odds ratios (AOR) and 95% confidence intervals (CI) were calculated. The statistical significance was set at $\leq$5%.

## 3. Results

Table 1 presents the proportion of participants who reported emotional distress during the first wave of the COVID-19 pandemic. There were 4515 (26.6%) participants who felt frustrated or bored; 4352 (25.8%) felt anxious, 2477 (14.7%) felt depressed, 2869 (17.0%) felt lonely, 1866 (11.1%) felt angry, and 1513 (9.0%) felt grief/a sense of loss. Among the 905 (5.4%) participants who reported living with HIV, 188 (20.8%) felt frustrated or bored, 238 (26.3%) felt anxious, 160 (17.7%) felt depressed, 148 (16.4%) felt lonely, 84 (9.3%) felt angry, and 53 (5.9%) felt grief/a sense of loss.

Table 2 shows that people living with HIV had significantly higher odds of feeling anxious (AOR: 1.644; 5% CI: 1.398–1.933; $p < 0.001$), depressed (AOR: 1.798; 95% CI: 1.489–2.171; $p < 0.001$), and lonely (AOR: 1.350; 95% CI: 1.115–1.635; $p = 0.002$) as compared to those living without HIV during the first wave of the COVID-19 pandemic.

Older age (AOR: 0.982), being male (AOR: 0.918) vs. female, and having no formal education (AOR: 0.305) or a primary level of education (AOR: 0.469), compared to a college/university education, was associated significantly with lower odds of frustration/boredom. Residents in LICs (AOR: 0.700) and LMICs (AOR: 0.561) compared to residents in HICs had significantly lower odds of frustration/boredom. Retired participants (AOR: 1.396) and students (AOR: 1.354) had significantly higher odds of frustration/boredom than those who were employed.

In addition, being male (AOR: 0.660), having no formal education vs. college/university education (AOR: 0.549), being unemployed vs. employed (AOR: 0.850), and living in LICs (AOR: 0.727) and LMICs (AOR: 0.452) vs. living in HICs was associated significantly with lower odds of feeling anxious.

Furthermore, older age (AOR: 0.978), being male (AOR: 0.822), and living in LICs (AOR: 0.649) and LMICs (AOR: 0.509) vs. living in HICs had significantly lower odds of feeling depressed. Respondents with a secondary level education (AOR: 1.162) vs. university/college education and unemployed respondents (AOR: 1.409) vs. employed respondents had significantly higher odds of feeling depressed.

Older age (AOR: 0.978), males (AOR: 0.874), having no formal education (AOR: 0.598), and living in LICs (AOR: 0.671), LMICs (AOR: 0.499), and UMICs (AOR: 0.700), compared to residents in HICs, was associated significantly with lower odds of feeling lonely during the pandemic. Respondents with a secondary level education (AOR: 1.157) vs. university/college education and respondents who were retired (AOR:1.489), students (AOR:1.337), or unemployed (AOR: 1.356), compared to those who were employed, had significantly higher odds of feeling lonely.

**Table 1.** Descriptive statistics of the dependent, independent, and confounding variables associated with HIV status in a multicountry sample of study participants (N = 16,866).

| Variables | Total | Frustration or Boredom | | Anxiety | | Depression | | Loneliness | | Anger | | Grief or Feeling of Loss | |
|---|---|---|---|---|---|---|---|---|---|---|---|---|---|
| | N = 16,866 n (%) | Yes 4515 (26.8) n (%) | No 12,351 (73.2) n (%) | Yes 4352 (25.8) n (%) | No 12,514 (74.2) n (%) | Yes 2477 (14.7) n (%) | No 14,389 (85.3) n (%) | Yes 2869 (17.0) n (%) | No 13,997 (83.0) n (%) | Yes 1866 (11.1) n (%) | No 15,000 (88.9) n (%) | Yes 1513 (9.0) n (%) | No 15,353 (91.0) n (%) |
| **Age mean (SD)** | 35.3 (12.9) | 33.1 (12.5) | 36.1 (13.0) | 35.7 (13.3) | 35.1 (12.8) | 32.8 (11.9) | 35.7 (13.0) | 32.4 (12.3) | 35.9 (13.0) | 34.2 (13.4) | 35.4 (12.8) | 34.4 (13.3) | 15,353 (12.9) |
| **Sex at birth** | | | | | | | | | | | | | |
| Male | 6366 (37.7) | 1575 (24.7) | 4791 (75.3) | 1322 (20.8) | 5044 (79.2) | 797 (12.5) | 5569 (87.5) | 959 (15.1) | 5407 (84.9) | 567 (8.9) | 5799 (91.1) | 428 (6.7) | 5938 (93.3) |
| Female | 10,500 (62.3) | 2940 (28.0) | 7560 (72.0) | 3030 (28.9) | 7470 (71.1) | 1680 (16.0) | 8820 (84.0) | 1910 (18.2) | 8590 (81.8) | 1299 (12.4) | 9201 (87.6) | 1085 (10.3) | 9415 (89.7) |
| **Educational level** | | | | | | | | | | | | | |
| None | 309 (1.8) | 25 (8.1) | 284 (91.9) | 37 (12.0) | 272 (88.0) | 40 (12.9) | 269 (87.1) | 31 (10.0) | 278 (90.0) | 14 (4.5) | 295 (95.5) | 13 (4.2) | 296 (95.8) |
| Primary | 398 (2.4) | 51 (12.8) | 347 (87.2) | 76 (19.1) | 322 (80.9) | 59 (14.8) | 339 (85.2) | 51 (12.8) | 347 (87.2) | 29 (7.3) | 369 (92.7) | 19 (4.8) | 379 (95.2) |
| Secondary | 2980 (17.7) | 940 (31.5) | 2040 (68.5) | 790 (26.5) | 2190 (73.5) | 537 (18.0) | 2443 (82.0) | 640 (21.5) | 2340 (78.5) | 423 (14.2) | 2557 (85.8) | 287 (9.6) | 2693 (90.4) |
| College/ university | 13,179 (78.1) | 3499 (26.5) | 9680 (73.5) | 3449 (26.2) | 9730 (73.8) | 1841 (14.0) | 11,338 (86.0) | 2147 (16.3) | 11,032 (83.7) | 1400 (10.6) | 11,779 (89.4) | 1194 (9.1) | 11,985 (90.9) |
| **Employment status** | | | | | | | | | | | | | |
| Retiree | 693 (4.1) | 166 (24.0) | 527 (76.0) | 201 (29.0) | 492 (71.0) | 72 (10.4) | 621 (89.6) | 94 (13.6) | 599 (86.4) | 90 (13.0) | 603 (87.0) | 72 (10.4) | 621 (89.6) |
| Student | 3750 (22.2) | 1374 (36.6) | 2376 (63.4) | 1030 (27.5) | 2720 (72.5) | 676 (18.0) | 3074 (82.0) | 885 (23.6) | 2865 (76.4) | 553 (14.7) | 3197 (85.3) | 440 (11.7) | 3310 (88.3) |
| Employed | 9787 (58.0) | 2338 (23.9) | 7449 (76.1) | 2531 (25.9) | 7256 (74.1) | 1245 (12.7) | 8542 (87.3) | 1389 (14.2) | 8398 (85.8) | 926 (9.5) | 8861 (90.5) | 759 (7.8) | 9028 (92.2) |
| Unemployed | 2636 (15.6) | 637 (24.2) | 1999 (75.8) | 590 (22.4) | 2046 (77.6) | 484 (18.4) | 2152 (81.6) | 501 (19.0) | 2135 (81.0) | 297 (11.3) | 2339 (88.7) | 242 (9.2) | 2394 (90.8) |
| **Country income level** | | | | | | | | | | | | | |
| LIC | 404 (2.4) | 98 (24.3) | 306 (75.7) | 106 (26.2) | 298 (73.8) | 50 (12.4) | 354 (87.6) | 64 (15.8) | 340 (84.2) | 28 (6.9) | 376 (93.1) | 44 (10.9) | 360 (89.1) |
| LMIC | 8935 (53.0) | 1978 (22.1) | 6957 (77.9) | 1666 (18.6) | 7269 (81.4) | 1057 (11.8) | 7878 (88.2) | 1275 (14.3) | 7660 (85.7) | 693 (693) | 8242 (92.2) | 537 (6.0) | 8398 (94.0) |
| UMIC | 3449 (20.4) | 1116 (32.4) | 2333 (67.6) | 1226 (35.5) | 2223 (64.5) | 636 (18.4) | 2813 (81.6) | 606 (17.6) | 2843 (82.4) | 591 (591) | 2858 (82.9) | 419 (12.1) | 3030 (87.9) |
| HIC | 4078 (24.2) | 1323 (32.4) | 2755 (67.6) | 1354 (33.2) | 2724 (66.8) | 734 (18.0) | 3344 (82.0) | 924 (22.7) | 3154 (77.3) | 554 (13.6) | 3524 (86.4) | 513 (12.6) | 3565 (87.4) |
| **Living with HIV** | | | | | | | | | | | | | |
| No | 15,961 (94.6) | 4327 (27.1) | 11,634 (72.9) | 4114 (25.8) | 11,847 (74.2) | 2317 (14.5) | 13,644 (85.5) | 2721 (17.0) | 13,240 (83.0) | 1782 (11.2) | 14,179 (88.8) | 1460 (9.1) | 14,501 (90.9) |
| Yes | 905 (5.4) | 188 (20.8) | 717 (79.2) | 238 (26.3) | 667 (73.7) | 160 (17.7) | 745 (82.3) | 148 (16.4) | 757 (83.6) | 84 (9.3) | 821 (90.7) | 53 (5.9) | 852 (94.1) |

**Table 2.** Binary logistic regression analysis determining the associations between emotional distress and HIV status of a multi-country sample of study participants (N = 16,866).

| Variables | Frustration or Boredom AoR (95% CI) (*p* Values) | Anxiety AoR; 95% CI (*p* Values) | Depression AoR; 95% CI (*p* Values) | Loneliness AoR; 95% CI (*p* Values) | Anger AoR; 95% CI (*p* Values) | Grief or Feeling of Loss AoR; 95% CI (*p* Values) |
|---|---|---|---|---|---|---|
| **Age** | 0.982; 0.979–0.986; $p < 0.001$ | 0.999; 0.995–1.003; $p = 0.582$ | 0.978; 0.973–0.983; $p < 0.001$ | 0.978; 0.974–0.983; $p < 0.001$ | 0.993; 0.988–0.998; $p = 0.006$ | 0.995; 0.989–1.000; $p = 0.055$ |
| **Sex at birth** | | | | | | |
| Male | 0.918; 0.853–0.988; $p = 0.023$ | 0.660; 0.612–0.712; $p < 0.001$ | 0.822; 0.749–0.903; $p < 0.001$ | 0.874; 0.801–0.953; $p = 0.002$ | 0.768; 0.691–0.854; $p < 0.001$ | 0.679; 0.603–0.765; $p < 0.001$ |
| Female | 1.000 | 1.000 | 1.000 | 1.000 | 1.000 | 1.000 |
| **Educational level** | | | | | | |
| No formal education | 0.305; 0.201–0.462; $p < 0.001$ | 0.549; 0.386–0.782; $p = 0.001$ | 0.928; 0.656–1.312; $p = 0.673$ | 0.598; 0.408–0.877; $p = 0.008$ | 0.512; 0.296–0.885; $p = 0.016$ | 0.576; 0.326–1.017; $p = 0.057$ |
| Primary | 0.469; 0.347–0.634; $p < 0.001$ | 0.808; 0.623–1.047; $p = 0.107$ | 1.128; 0.846–1.503; $p = 0.413$ | 0.802; 0.592–1.085; $p = 0.153$ | 0.775; 0.526–1.143; $p = 0.198$ | 0.614; 0.383–0.982; $p = 0.042$ |
| Secondary | 1.086; 0.990–1.192; $p = 0.080$ | 0.979; 0.889–1.078; $p = 0.665$ | 1.162; 1.038–1.301; $p = 0.009$ | 1.157; 1.041–1.286; $p = 0.007$ | 1.215; 1.073–1.376; $p = 0.002$ | 0.941; 0.815–1.086; 0.404 |
| College/university | 1.000 | 1.000 | 1.000 | 1.000 | 1.000 | 1.000 |
| **Employment status** | | | | | | |
| Retiree | 1.354; 1.100–1.666; $p = 0.004$ | 0.934; 0.766–1.137; $p = 0.495$ | 1.125; 0.849–1.490; $p = 0.413$ | 1.489; 1.153–1.923; $p = 0.002$ | 1.267; 0.967–1.659; $p = 0.086$ | 1.313; 0.976–1.765; $p = 0.072$ |
| Student | 1.396; 1.262–1.545; $p < 0.001$ | 1.054; 0.949–1.171; $p = 0.328$ | 1.068; 0.942–1.212; $p = 0.305$ | 1.337; 1.189–1.503; $p < 0.001$ | 1.393; 1.210–1.605; $p < 0.001$ | 1.452; 1.245–1.694; $p < 0.001$ |
| Employed | 1.000 | 1.000 | 1.000 | 1.000 | 1.000 | 1.000 |
| Unemployed | 1.037; 0.932–1.153; $p = 0.508$ | 0.850; 0.762–0.947; $p = 0.003$ | 1.409; 1.247–1.592; $p < 0.001$ | 1.356; 1.203–1.527; $p < 0.001$ | 1.221; 1.056–1.412; $p = 0.007$ | 1.259; 1.074–1.476; $p = 0.004$ |
| **Country income level** | | | | | | |
| LICs | 0.700; 0.551–0.889; $p = 0.003$ | 0.727; 0.576–0.917; $p = 0.007$ | 0.649; 0.476–0.884; $p = 0.006$ | 0.671; 0.507–0.888; $p = 0.005$ | 0.503; 0.339–0.747; $p = 0.001$ | 0.888; 0.639–1.233; $p = 0.478$ |
| LMICs | 0.561; 0.514–0.611; $p < 0.001$ | 0.452; 0.414–0.493; $p < 0.001$ | 0.509; 0.457–0.568; $p < 0.001$ | 0.499; 0.452–0.551; $p < 0.001$ | 0.509; 0.449–0.576; $p < 0.001$ | 0.437; 0.382–0.498; $p < 0.001$ |
| UMICs | 0.977; 0.885–1.078; $p = 0.645$ | 1.088; 0.988–1.198; $p = 0.087$ | 1.010; 0.897–1.138; $p = 0.869$ | 0.700; 0.623–0.786; $p < 0.001$ | 1.273; 1.121–1.446; $p < 0.001$ | 0.918; 0.799–1.056; $p = 0.232$ |
| HICs | 1.000 | 1.000 | 1.000 | 1.000 | 1.000 | 1.000 |
| **Living with HIV** | | | | | | |
| No | 1.000 | 1.000 | 1.000 | 1.000 | 1.000 | 1.000 |
| Yes | 1.068; 0.898–1.269; $p = 0.459$ | 1.644; 1.398–1.933; $p < 0.001$ | 1.798; 1.489–2.171; $p < 0.001$ | 1.350; 1.115–1.635; $p = 0.002$ | 1.267; 0.995–1.615; $p = 0.055$ | 1.025; 0.764–1.376; $p = 0.869$ |

Older age (AOR: 0.993), being male (AOR: 0.768), having no formal education (AOR: 0.512) vs. college/university education, and living in LICs (AOR: 0.503) and LMICs (AOR: 0.509), compared to HICs, was significantly associated with lower odds of feeling angry during the pandemic. Respondents with secondary level of education (AOR: 1.215) vs. college/university education had significantly higher odds of feeling angry. Students (AOR: 1.393) and those who were unemployed (1.221), compared to those who were employed, and respondents from UMICs (AOR: 1.273) vs. HICs had significantly higher odds of feeling angry.

With respect to feeling grief or loss, males (AOR: 0.679), respondents with a primary level of education vs. college/university education (AOR: 0.614), and respondents from LMICs vs. HICs (AOR: 0.437) had significantly lower odds of feeling grief/a sense of loss. However, students (AOR: 1.452) and those who were unemployed (AOR:1.259) had significantly higher odds of feeling grief/a sense of loss.

## 4. Discussion

This manuscript reports on a study that explored a host of multidimensional factors thought to be associated with different forms of emotional distress experienced during the first wave of the COVID-19 pandemic. This study signals that people living with HIV are more likely to feel all forms of emotional distress as compared to people living without HIV during the first wave of the COVID-19 pandemic. The differences between these two populations in this study were significant for anxiety, depression, and loneliness factors. Sociodemographic factors were also related to different forms of emotional distress. The present findings also indicated that the odds of emotional distress decreased with age, significantly so for frustration/boredom, depression, loneliness, and anger. Sex differences were observed whereby men were less likely to report any form of emotional distress as compared to women. Individuals from low- and low–middle-incomes were found significantly less likely to feel any form of emotional distress as compared to those with higher incomes. Moreover, education level was a significant factor in predicting emotional distress, and the current findings pointed to those without formal education as being less likely to experience emotional distress. Furthermore, employment status is implicated in emotional distress, whereby individuals who were unemployed during the pandemic were more likely to report feeling depression, loneliness, anger, and grief but less anxiety. Finally, students reported feeling frustrated/bored, lonely, angry, and a sense of loss, whilst retired persons were more likely to feel frustrated/bored and lonely.

To expand, it was observed that the COVID-19 pandemic introduced additional layers of stress, particularly for those belonging to a special population. Indeed, as supported by this study and earlier works, COVID-19 appeared to amplify various forms of distress, such as anxiety [51], depression [51], and loneliness [2], for people living with HIV. People living with HIV are a group who are constantly living with or facing triggers for emotional distress, including food insecurity [30,52–56], financial stress [31], comorbid chronic medical condition [57], and significant lifestyle changes, which are associated with a new diagnosis of HIV [58]. Prior research has indicated that a significant number of people living with HIV suffered emotional distress during the pandemic [59]. The present study confirms that emotional distress can be worse for people living with HIV than those living without HIV, especially during a pandemic. These findings should motivate decision-makers to provide additional mental health care support for people living with HIV during pandemic times, even whilst receiving standard HIV care provisions.

Consideration of the ways in which the COVID-19 pandemic impacts emotional health for different sociodemographic groups introduces new complexities to the observed relationship between HIV status and emotional health. Age, sex, educational status, and employment status were all associated with emotional distress. These factors may also have mediated the associations between HIV status and emotional distress witnessed during the first wave of the COVID-19 pandemic. Younger people have been shown to have an increased propensity to mental health conditions during the COVID-19 pandemic [60–63],

and this may help to explain why students in the current study were more likely to experience emotional distress. Emotional distress among young people during the pandemic is linked to isolation from friends and social support networks. These concerns are further perpetuated when issues such as increased time spent on social media and other online activities, worries over household finances, COVID-19-related illness, a reduction in sports and cultural activities, and watching mainstream media news are considered [64,65].

Moreover, as supported by the current findings, women are more likely to report emotional distress during a pandemic [26,66,67], although it is possible that the reasons behind this association stem from gender differences in the willingness to report distress, tolerance for frustration, stress responsivity, and other biological, environmental, and social factors [68,69]. In contrast, people with no formal or primary-level education are less likely than those with college/university education to experience any form of emotional distress. This could be because people with lower educational status were more likely to experience the precipitating factors for emotional stress prior to the pandemic [70–72] and may be less likely to experience worsening distress from the shocks of the pandemic. On the contrary, people with higher educational status may be more likely to experience a greater degree of emotional distress despite having weathered the pandemic under more favorable conditions and privileged economic circumstances [73].

A key strength of the findings reported in the current paper is that they are derived from a study that benefited from a large sample, which enabled a robust sub-analysis to be conducted. Additionally, the multi-country dataset allows for the generalizability and representation of these findings on a global scale. The study also contributes new information to the public understanding of the emotional impact of the pandemic on different groups of people. Similar to all cross-sectional studies, it was not, however, possible to establish clear causal pathways from the current data. It is acknowledged that non-probability sampling of the online participants may have skewed the data to those with a higher educational level and those more likely to have access to the internet. Upon reflection, using an online survey data collection method was the most appropriate choice in this scenario due to the distancing and lockdown restrictions in place during the time that this study was carried out [74–76]. The current findings provide fruitful thinking and further hypotheses to be tested by future research.

## 5. Conclusions

The associations between emotional distress and HIV status identified by this study suggest that people living with HIV experienced significant emotional distress during the first wave of the COVID-19 pandemic. The COVID-19 pandemic can exacerbate the emotional stress for those living with HIV, and the system in which COVID-19 impacts emotional health among different sociodemographic groups introduces further complexities regarding this observed effect. Greater attempts to offer extra mental health care support for people living with HIV during pandemic periods are encouraged.

**Author Contributions:** Conceptualization, M.O.F. and J.L.; methodology, M.O.F. and A.L.N.; validation, M.O.F. and A.L.N.; formal analysis, R.A.A.Z.; data curation, N.M.A.; writing—original draft preparation, M.O.F.; writing—review and editing, M.O.F., R.A.A.Z., J.I.V., P.E., M.A.Y., B.E.O., B.G., F.B.L., Z.K., N.M.A., J.L. and N.M.A.; supervision, M.O.F.; project administration, M.O.F., A.L.N. and N.M.A. All authors have read and agreed to the published version of the manuscript.

**Funding:** This research received no external funding.

**Institutional Review Board Statement:** Ethical approval was obtained from the Human Research Ethics Committee at the Institute of Public Health of the Obafemi Awolowo University Ile-Ife, Nigeria (HREC No: IPHOAU/12/1557), Brazil (CAAE N° 38423820.2.0000.0010), India (D-1791-uz and D-1790-uz), Saudi Arabia (CODJU-2006F), and the United Kingdom (13283/10570).

**Informed Consent Statement:** Informed consent was obtained from all participants involved in this study.

**Data Availability Statement:** Data is contained within the article.

**Acknowledgments:** The authors wish to thank those who participated in this study.

**Conflicts of Interest:** The authors declare no conflict of interest.

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
