# Peer review of "The Impact of COVID-19 on the Emotion of People Living with and without HIV"

_2673-947X, doi:10.3390/hygiene3010005_

Round 1

Reviewer 1 Report

After reading this article, I noticed the following aspects related to:

Abstract. The authors specify the general objective of their research, the methodology used, the results and conclusions of their research.

Introduction. This section is condensed and does not specify many aspects related to gap coverage in the literature. I suggest the authors to clearly specify the mentioned. Also, there is no literature review part and no clear subsections delimiting the investigated aspects of the authors' research.

Materials and methods. This section specifies aspects of the research methodology, sampling and method of development, accompanied by the analysis of the data and interview participants.

Results. The authors specify the results obtained after carrying out their research. The data in the tables are unclear and I suggest the authors to enlarge them or find a way to make them more visible to the readers.

Discussions. This section interprets the results of the research and ends with some conclusions that should have been moved to a separate section of Conclusions, research limits and future research directions. I suggest the authors to realize these aspects.

Author Response

Thank you for your helpful comments, our responses are below:

Abstract. The authors specify the general objective of their research, the methodology used, the results and conclusions of their research.

Response: Thanks for the positive feedback

Introduction. This section is condensed and does not specify many aspects related to gap coverage in the literature. I suggest the authors to clearly specify the mentioned. Also, there is no literature review part and no clear subsections delimiting the investigated aspects of the authors' research.

Response: We agree with the reviewer and have extensively revised the introduction. We feel the revised introduction would have addressed the concerns of the reviewer.

Materials and methods. This section specifies aspects of the research methodology, sampling and method of development, accompanied by the analysis of the data and interview participants.

Response: Thanks for the positive feedback

Results. The authors specify the results obtained after carrying out their research. The data in the tables are unclear and I suggest the authors to enlarge them or find a way to make them more visible to the readers.

Response: We have increased the font of the results

Discussions. This section interprets the results of the research and ends with some conclusions that should have been moved to a separate section of Conclusions, research limits and future research directions. I suggest the authors to realize these aspects.

Response: we have created a sub-heading known as conclusion

Reviewer 2 Report

The impact analysis of COVID-19 on emotion among on emotion among those living with HIV and without HIV is presented in the manuscript. The presented results are interesting, but the manuscript needs improvement. I suggest you study the editorial guidelines and adapt the article carefully.

Moreover:

The abstract is too long, according to the guidelines - max 200 words.

Introdution: The introduction should be developed, in particular, why it was decided to compare the emotional state of people with HIV and without HIV during the pandemic. The following explanation is not sufficient:

67-69 ,,However, it has recently been argued that the emotional responses elicited by the pandemic have a lower impact on people living with HIV because they are better prepared 68 for this type of emotionally stressful event [33, 34].”

Material and methods:

In my opinion, however, the exclusion criteria should be specified - age under 18

Results

It is better to start the text describing the results in the table with a new paragraph, and attach the reference to which the table refers to - before the full stop at the end of the sentence.152-156 and 189-191 should be indented according to the journal's guidelines.

Table descriptions - editorial guidelines of the work were not applied

In Table 1, in the Results section-  N=16,866 is given, but in the Material and methods section- N=21,106. Over 4,000 respondents were excluded? For what reason ? It should be clearly stated in the material and methods section why 16866 was analyzed and over 4000 were excluded.

The impact analysis of COVID-19 on emotion among on emotion among those living with HIV and without HIV is presented in the manuscript. The presented results are interesting, but the manuscript needs improvement. I suggest you study the editorial guidelines and adapt the article carefully.

Moreover:

The abstract is too long, according to the guidelines - max 200 words.

Introdution: The introduction should be developed, in particular, why it was decided to compare the emotional state of people with HIV and without HIV during the pandemic. The following explanation is not sufficient:

67-69 ,,However, it has recently been argued that the emotional responses elicited by the pandemic have a lower impact on people living with HIV because they are better prepared 68 for this type of emotionally stressful event [33, 34].”

Material and methods:

In my opinion, however, the exclusion criteria should be specified - age under 18

Results

It is better to start the text describing the results in the table with a new paragraph, and attach the reference to which the table refers to - before the full stop at the end of the sentence.152-156 and 189-191 should be indented according to the journal's guidelines.

Table descriptions - editorial guidelines of the work were not applied

In Table 1, in the Results section-  N=16,866 is given, but in the Material and methods section- N=21,106. Over 4,000 respondents were excluded? For what reason ? It should be clearly stated in the material and methods section why 16866 was analyzed and over 4000 were excluded.

Author Response

Thank you for your helpful comments, our responses are below:

The impact analysis of COVID-19 on emotion among those living with HIV and without HIV is presented in the manuscript. The presented results are interesting, but the manuscript needs improvement. I suggest you study the editorial guidelines and adapt the article carefully.

Response: we have carefully addressed the issues raised and hope this has improved the quality of the manuscript

Moreover:

The abstract is too long, according to the guidelines - max 200 words.

Response. The words have been reduced to 200

Introdution: The introduction should be developed, in particular, why it was decided to compare the emotional state of people with HIV and without HIV during the pandemic. The following explanation is not sufficient: 67-69 ,,However, it has recently been argued that the emotional responses elicited by the pandemic have a lower impact on people living with HIV because they are better prepared 68 for this type of emotionally stressful event [33, 34].”

Response: Thanks for highlighting this. We have revised the introduction extensively

Material and methods:

In my opinion, however, the exclusion criteria should be specified - age under 18

Response: The inclusion criteria were those 18 years and above. The study had no exclusion criteria

Results

It is better to start the text describing the results in the table with a new paragraph, and attach the reference to which the table refers to - before the full stop at the end of the sentence.152-156 and 189-191 should be indented according to the journal's guidelines.

Response. We have indented all the paragraphs. However, there are only 2 tables and we made reference to the tables at the beginning of the paragraph reporting the tables

Table descriptions - editorial guidelines of the work were not applied. In Table 1, in the Results section-  N=16,866 is given, but in the Material and methods section- N=21,106. Over 4,000 respondents were excluded? For what reason? It should be clearly stated in the material and methods section why 16866 was analyzed and over 4000 were excluded.

Response. Thanks for pointing out this disparity in figures. We have included details on the study design. We note that: This was a secondary analysis of data extracted from a primary study on mental health and wellness. We extracted the data of 16,866 participants that had complete variables, for the study.

The impact analysis of COVID-19 on emotion among on emotion among those living with HIV and without HIV is presented in the manuscript. The presented results are interesting, but the manuscript needs improvement. I suggest you study the editorial guidelines and adapt the article carefully.

Response. We hope the revisions made has addressed the concerns.

Reviewer 3 Report

Dear Authors, 

I read with interest your work, as people suffering from chronic diseases need more attention on the way they experienced the pandemic. I liked also your focus on HIV and emotional burden.

In general, I think that your work has almost all the necessary elements to be published, but it seems to me too compressed and synthetic, especially in the Introduction. 

I would suggest to expand the Introduction by analyzing with more detail the impact of COVID-19 on mental health and on other contexts of life (education, work, social relationships). 

I appreciated, instead, the statistical analyses and the Discussion for their degree of thoroughness. 

I support the publication of the manuscript after a minor revision of the Introduction.

Author Response

Dear Reviewer, thank you for your helpful feedback, our comments are below:  

I read with interest your work, as people suffering from chronic diseases need more attention on the way they experienced the pandemic. I liked also your focus on HIV and emotional burden.

Response. Thanks for the positive feedback

In general, I think that your work has almost all the necessary elements to be published, but it seems to me too compressed and synthetic, especially in the Introduction. I would suggest to expand the Introduction by analyzing with more detail the impact of COVID-19 on mental health and on other contexts of life (education, work, social relationships). 

Response. We agree with this observation. We have improved on the content and the length of the paper. We hope the edits addresses the concerns of the reviewer

I appreciated, instead, the statistical analyses and the Discussion for their degree of thoroughness. 

I support the publication of the manuscript after a minor revision of the Introduction.

Response. Thanks for the positive feedback

Round 2

Reviewer 1 Report

Thanks to the authors for making the indicated suggestions.

Author Response

Thank you for taking the time to review our manuscript. It is very much appreciated